# Optimisation of Potato Dextrose Agar Culture Medium for Actinobacteria Growth

**DOI:** 10.3390/microorganisms13030654

**Published:** 2025-03-13

**Authors:** Elian Chaves Ribeiro, Emanuelle Ketthlen Nunes Araújo, Margareth Santos Costa Penha, Adriana Silva do Nascimento, Darlan Ferreira da Silva, Rita de Cássia Mendonça de Miranda

**Affiliations:** 1Postgraduate Department, Postgraduate Program in Bioscience Applied to Health, Ceuma University, São Luis 65075-120, MA, Brazil; elian.chaves@ufma.br (E.C.R.); emanuelleketthlen@gmail.com (E.K.N.A.); margareth.penha@ufma.br (M.S.C.P.); 2Postgraduate Department, Federal Institute of Maranhão, São Luis 65075-441, MS, Brazil; adriananascimento@ifma.edu.br; 3Postgraduate Department, Postgraduate Program in Environment, Ceuma University, São Luis 65075-120, MA, Brazil; darlan005229@ceuma.com.br

**Keywords:** optimisation, soil, secondary metabolites, *Streptomyces*, antimicrobial

## Abstract

The objective of this study was to optimise the potato dextrose agar (PDA) culture medium in terms of its potential for use in the growth of actinobacteria. The strain used in this study was a species of actinobacteria previously identified as *Streptosporangium* sp. (P1C3), characterised by slow growth (20 days of incubation), low aerial mycelium production, and no pigment production. To determine the optimal formulation, the *Streptosporangium* sp. (P1C3) strain was tested for incubation time and aerial mycelium growth across 27 formulations based on the PDA culture medium. A central composite rotational design (CCRD) experimental methodology was employed, where glucose concentration (g/L), yeast extract concentration (g/L), pH, and temperature were tested. Among the tested formulations, 01, 05, 09, and 13 showed a reduction in incubation time and complete aerial mycelium growth, which was linearly influenced by the four tested variables. Response surface analysis indicated that the optimal values for promoting aerial mycelium growth in the shortest incubation time were 10 g/L glucose concentration, from 1 g/L to 3 g/L yeast extract concentration, pH levels between 5.7 and 7.2, and temperatures between 24 °C and 32 °C. The optimisation of the PDA medium proved effective in improving the isolation of actinobacteria and enhancing the production of metabolites with potential antimicrobial activity.

## 1. Introduction

Soil is a naturally diverse ecosystem where complex biological communities develop, harbouring a variety of microorganisms, both eukaryotic and prokaryotic, that coexist in an environment subject to constant changes, interacting dynamically and in balance. The growth and evolution of these organisms are influenced by various factors, such as the presence of organic substrates and abiotic conditions (temperature, humidity, and aeration) [1].

Microorganisms, in general, play a crucial role in decomposing organic residues and generating humic substances that aggregate primary particles and form soil aggregates. This activity also promotes nutrient cycling, making them available to plants. Moreover, most of these microorganisms produce a wide range of bioactive compounds through their efficient metabolic activity, which plays a significant role in combating diseases [2], among which actinobacteria stand out.

Actinobacteria are Gram-positive, obligate aerobic, chemoorganotrophic, and non-motile bacteria, characterised by their filamentous structure and high guanine-cytosine (G+C) content in their DNA, approximately 78% [3,4]. These bacteria are important producers of secondary metabolites with antibacterial, antifungal, antitumour, antiprotozoal, and antiviral properties, in addition to vitamins and enzymes. Approximately 75% of known antibiotics are derived from actinobacteria, with two-thirds of these compounds produced by species of the genus *Streptomyces* [5,6].

Despite the numerous benefits of metabolites derived from actinobacteria, the isolation and cultivation of these microorganisms are considered costly processes for research laboratories due to the high costs of appropriate culture media, which have highly diverse compositions. Several formulations with different nutrient bases and carbon sources are used for the isolation of these bacteria, including Gause’s agar medium; arginine glycerol salt medium; Czapek agar medium; Bennett’s agar; humic acid-vitamin agar medium; and starch casein nitrate agar medium [7], according to Sharma et al. (2014) [8]. However, there are no similarities between these media, as most have varied compositions. Therefore, the need for alternative formulations for the isolation of actinobacteria using more accessible and cost-effective components has motivated the scientific community to explore alternative formulations.

Potato dextrose agar (PDA) is widely used to promote microbial growth and facilitate the isolation of individual strains [1,2,6]. This medium provides dextrose as a carbon source and essential nutrients derived from potato extract, creating suitable conditions for the development of actinobacteria. This formulation allows for the macroscopic observation of colonies, aiding their characterisation. The hypothesis suggests that the optimised culture medium will promote the growth of the mycelium of *Streptosporangium* sp. PC3 in a shorter incubation time, accelerating the process of obtaining the metabolite. In this context, the present study aims to optimise the potato dextrose agar culture medium for the growth of *Streptosporangium* sp. (P1C3) using an experimental design methodology.

## 2. Materials and Methods

### 2.1. Microorganisms

For the optimisation of the cultivation medium, the isolate previously identified as *Streptosporangium* sp. (PC3) was used [9]. This strain was isolated from beach sand and is characterised by slow growth and the absence of pigment when grown on organic PDA for up to 15 days of incubation at 30 °C. The bacterial strains belong to the culture collection of the Biotechnology and Electrochemistry Laboratory of Ceuma University and are registered with the code BIOTECP1C3.

### 2.2. Obtaining and Optimising the Formulation for the Growth of Streptosporangium sp.

To determine the treatment conditions, a fractional factorial experimental design matrix methodology was employed, consisting of two levels (−1, +1), two axial points (−2.12, +2.12), and three central points, totaling 27 experiments. The independent variables were X_1_—glucose concentration (g/L), X_2_—yeast extract concentration (g/L), X_3_—pH, and X_4_—temperature (°C), while the dependent variables were incubation time and aerial mycelium growth (Table 1). The experimental matrix and analysis of the results were processed using Statistic version 8.0 software [10].

Twenty-seven formulations were prepared in Petri dishes, based on organic PDA, with variations in nutrients, incubation time, and temperature, according to the experimental design. Of these formulations, 16 combined two levels (−1 and +1), 8 used axial levels (−2.12 and +2.12), and 3 were central points. All formulations were sterilised at 121 °C and supplemented with Amphotericin B (0.5%). To evaluate the efficacy of the formulations, mycelial growth was analysed at different time points, with codes assigned as follows: 0 (no growth), 5 (moderate growth), and 10 (full growth). In addition to growth time, the presence or absence of pigment production was also assessed. A response surface methodology was used to establish the optimised conditions based on the predicted model after analysis of variance of the experimental design results. In addition to the growth time, the presence or absence of pigment production was also assessed [10].

### 2.3. Scanning Electron Microscopy (SEM) Analysis Coupled with EDS (Energy Dispersive Spectroscopy)

For the observation of the growth and micromorphology of *Streptosporangium* sp. spores grown in the best formulations, the scanning electron microscopy (SEM) technique was used. This technique involves applying a finely focused electron beam that scans the surface of a specific area of the sample, enabling high-resolution morphological analysis and generating three-dimensional images magnified up to 100,000 times [11,12,13].

Additionally, energy dispersive spectroscopy (EDS), coupled with SEM, was used to map the elements present in a specific area of the sample, providing a detailed amplification of the structures. The method requires the sample to be solidified due to the internal vacuum of the equipment [13], and for this purpose, the samples were freeze-dried using high-precision equipment from the Solab brand.

### 2.4. Statiscal Analyses

The significant variables in the shaping process were assessed using the ANOVA methodology, considering a 95% confidence level (*p* < 0.05) in Origin 8.0 software.

## 3. Results

### Obtaining and Optimising the Formulation for the Growth of Actinobacteria

The formulation for the growth of *Streptosporangium* sp. (P1C3) was obtained and optimised using a factorial design methodology with response surface plots, employing Origin software (version 8.0).

The values of the independent variables, as well as the results obtained for time and mycelium growth after inoculation of *Streptosporangium* sp. in the 27 formulations predicted by the experimental design, can be seen in Figure 1. After completing the 27 experiments, it was observed that *Streptosporangium* sp. grew well within seven days of incubation in formulations 05, 09, and 13, while in formulation 01, it grew well after ten days of incubation. In the other formulations, *Streptosporangium* sp. exhibited slower growth, with fifteen to twenty days of incubation (Figure 1A). Although the amounts of carbon and nitrogen sources were the same in the experiments (10 g/L of glucose and 1 g/L of yeast extract), there were variations in pH (7.2–5.7) and incubation temperature (24 °C–32 °C). When the response variable was the development of aerial mycelium, it was observed that formulations 01, 05, 09, 11, 13, and 15 were the most efficient, as they exhibited total growth of aerial mycelium (Figure 1B). In this case, only glucose remained constant across the top five formulations, with variations in the concentrations of yeast extract, pH, and temperature.

In the experiments conducted, a reduction in incubation time from 20 to 7 days was observed, with total mycelium growth in the three formulations, indicating that, despite variations in temperature and pH, nutrients were the main factors influencing the accelerated growth of aerial mycelium.

Analysis of the results by ANOVA revealed that the four independent variables, in their linear form, significantly impacted the process of obtaining the best formulations, with *p* < 0.05 for incubation time and mycelial growth.

Additionally, the interaction between glucose concentration/pH and glucose/temperature also had a linear effect on incubation time and mycelial growth, respectively, with *p* < 0.05 (Table 2).

To generate the response surface plots from the ANOVA (Figure 2, the predicted models for the variables incubation time and mycelial growth are represented in Equations (1) and (2).Incubation time = 21.4 + 18.6*x*^1^ + 8.6*x*^2^ + 5.7*x*^3^ + 7.8*x*^4^ + 8.5*x*^1^*x*^3^.(1)Mycelial growth = 98.9 + 54.8*x*^1^ + 3.8*x*^2^ + 2.4*x*^3^ + 3.1*x*^4^ + 419.7*x*
^1^*x*^4^.(2)

Based on the predicted models, we generated response surface graphs to determine the trends of the adsorbent values and the time for adsorbate treatment (Figure 3).

Although the four variables influenced the incubation time and aerial mycelium growth in a linear manner, the response surface graphs showed some differences.

When the dependent variable was mycelium growth, the response surface was generated from the glucose concentration and incubation temperature variables, and what was observed was a positive trend with increasing values for both variables. The response surface graph points to a 10 g/L glucose concentration as the optimal value, while for temperature, the optimal values show an increasing trend, ranging from 24 °C to 32 °C (Figure 3A).

When the dependent variable was incubation time, the response surface was generated from the glucose concentration and pH variables, and a negative trend with decreasing values for both variables was observed. The response surface graph shows that a 10 g/L glucose concentration is the optimal value, while for pH, the optimal values show a decreasing trend, ranging from 7.2 to 5.7 (Figure 3B).

From the values shown in the response surface graph, it can be suggested that a 10 g/L glucose concentration, 1 g/L–3 g/L yeast extract concentration, pH values between 7.2 and 5.7, and temperatures between 24 °C and 32 °C would be the optimal values to work with these microorganisms to obtain aerial mycelium growth in the shortest incubation time.

The response surface methodology consists of a collection of statistical and mathematical techniques useful for developing, improving, and optimising treatment processes.

In this work, the methodology was used to optimise the growth time of *Streptosporangium* sp., thereby reducing the incubation time required for the development of aerial mycelium.

Another aspect observed was the morphology of *Streptosporangium* sp. in the 27 formulations. After being incubated at the temperature and time determined in the experimental design, the growth, aerial mycelium formation, and presence of pigment were evaluated. The macromorphological differences in *Streptosporangium* sp. used in the experimental design can be seen in Figure 2.

Of the 27 formulations developed based on the experimental method, formulations 01, 05, 09, and 13 showed a reduction in mycelial growth and incubation time. Previously, the *Streptosporagium* sp. (P1C3) strain required 20 days of incubation for mycelium development. After 7 days of incubation in an incubator, the four formulations (01, 05, 09, 13) exhibited mycelial growth, similar to the commercial formulation, with colonies appearing dry, with a blackish-green colour, and a firm consistency.

Another important aspect to mention is that the macroscopic observation of formulations 01, 05, 09, and 13 enabled the detection of pigment, suggesting that *Streptosporagium* sp. secretes pigment externally into the medium.

For the microscopic observation, *Streptosporagium* sp. (P1C3) grown in the promising formulations identified in the experimental method (01, 05, 09, and 13), as well as the commercial formulation, was subjected to scanning electron microscopy (SEM) coupled with energy-dispersive X-ray spectroscopy (EDS). The *Streptosporagium* sp. (P1C3) strain displayed distinct characteristics among the experimental and commercial formulations.

In formulation 01, the *Streptosporagium* sp. strain showed a tangle of hyphae with low spore organisation at the ends, which is important for the formation of aerial vegetative mycelium. This formulation allowed sporulation in a lower quantity of actinobacteria, resulting in limited formation of aerial mycelium. In contrast, the *Streptosporagium* sp. (P1C3) strain grown in experimental formulation 5 demonstrated intense hyphal formation and a large tangle of spores, favouring the development of aerial mycelium. This indicates that formulation 5 promoted more abundant sporulation of actinobacteria, directly affecting the formation of aerial mycelium.

The morphology of the *Streptosporagium* sp. strains cultivated in the four experimental formulations and the commercial formulation (PDA Difco^@^) can be more clearly observed in Figure 4.

Energy dispersive spectroscopy (EDS) was applied to formulations 01, 05, 09, and 13 with growth of the *Streptosporagium* sp. strain, and to the commercial formulation of PDA for a detailed analysis of the chemical composition. The results from formulations 05, 09, and 13 indicated no difference in composition. The elements C, O, K, and P were present in the composition; however, it was observed that in formulations 05, 09, and 13, the elements K and P showed a higher chemical percentage compared to the commercial formulation (Table 3).

When formulation 01 was analysed, the presence of two distinct elements, Al and In (aluminium and indium), was noted, which were not present in the other experimental samples or the commercial formulation. This may explain the increased incubation time when *Streptosporangium* sp. was cultivated in this formulation, as shown in Table 3.

When comparing the formulations with the appearance of morphological structures such as hyphae and spores, it was observed that in formulation 09, there was abundant production of free and germinative spores, whereas the appearance of hyphae was discreet in comparison with the other formulations that directly interfered with the reproduction of the aerial hyphae and, consequently, with the formation of aerial mycelium.

Formulation 13 stood out due to the abundant formation of aerial hyphae in a spiral structure, allowing the visibility of the terminal structures of the hyphae, that is, the aerial mycelium. Furthermore, formulation 13 proved to be favourable for sporulation.

The commercial formulation of PDA displayed distinct characteristics compared to the four experimental formulations tested. In the commercial formulation, there was intense formation of reproductive hyphae, along with a small number of spherical spores and aerial hyphae. However, septations or maturation at the tips of the hyphae were not observed, processes that are essential for the formation of aerial mycelium, which is crucial for the production of pigments, secondary metabolites, and bioactive compounds, such as potential antimicrobial and antiparasitic agents, among others.

## 4. Discussion

Based on the original formulation of the PDA containing 20 g/L of glucose and 2 g/L of yeast extract, a comparative study was conducted with 27 formulations generated from a central composite rotational (CCR) experimental design. It was observed that formulation 15, containing 10 g/L of glucose, 1 g/L of yeast extract, and with a pH of 7.25, showed colony growth after 7 days. However, aerial mycelium formation occurred only after 10 days of incubation at 32 °C. In formulation 1, where 10 g/L of glucose, 1 g/L of yeast extract, a temperature of 24 °C, and a pH of 5.75 were used, the colony exhibited growth and aerial mycelium formation after 7 days of incubation.

Given this, it was observed that variations in media and components can influence incubation time, mycelial growth, and the morphology of *Streptosporangium* sp. strains. This study confirmed this observation, as the reduction in the constituents of the medium used influenced the growth of the *Streptosporangium* sp. P1C3 strain, which grew at a pH ranging from 5.75 to 7.25 and temperatures between 24 °C and 32 °C.

The bioactive compounds produced by actinobacteria are often enhanced through the optimisation of growth conditions. In some cases, significant improvements can be achieved by increasing or decreasing the oxygen transfer rate, altering the carbon source, and regulating the availability of nitrogen and phosphate [14,15].

In a study conducted by Lertcanawanichakul and Sahabuddee [16], variations in the morphology and physiology of *Streptomyces* sp. KB1 were observed by optimising solid medium conditions, altering its components, and performing analyses in different culture media through biochemical tests. Changes in pH, temperature, the addition of carbon and nitrogen sources, NaCl, and trace elements resulted in favourable effects on the mycelial growth of the microorganism over 4 days, similar to what was observed in this study [16].

Bioactive compounds produced by *Streptomyces* strains are often enhanced through the optimisation of growth conditions. In some cases, significant improvements can be achieved by increasing or decreasing the oxygen transfer rate, altering the carbon source, and regulating the availability of nitrogen and phosphate [15]. This can be better understood when the concentration of glucose and yeast extract in the PDA medium is increased, compared to formulation 1, where it was observed that in a very concentrated medium, it negatively affected the incubation time, preventing the growth of aerial mycelium.

In formulation 8, 30 g/L of glucose and 2 g/L of yeast extract were used, which resulted in an increase in incubation time to 20 days, i.e., double the time, during which mycelial growth did not occur. The same effect was observed when the yeast extract concentration tripled, keeping the glucose concentration at 30 g and reducing the pH to 5.75, which also led to an increase in incubation days and the absence of mycelial growth.

Another important feature of the colonies is their morphological diversity, primarily based on reproductive strategies, leading to the formation of a variety of spore structures [16], where colonies typically show slow growth, requiring around seven to ten days to develop their aerial hyphae [17]. Escher [18] reported in his study that most isolates developed aerial mycelium after five days of cultivation in ISP2 medium. In the presence of PDA medium, a medium with minimal growth for the isolated actinobacterial strains, growth was only recorded for the MPO3 strain [18].

In a situation where one of the components of the culture medium is excluded, it can also influence mycelial growth time. This phenomenon can be observed in the formulation where only 20 g/L of glucose is present, at a pH of 6.5 and a temperature of 28 °C. In this formulation, 15 days were needed for colony growth of the strain used in the study, and an additional 5 days for mycelial growth.

The pH can also significantly influence the growth and isolation of colonies, as, for most microorganisms in this group, a pH close to neutrality is optimal, with few exceptions that sporulate only in the pH range between 4.5 and 5.0 [19]. Streptomycetaceae bacterial groups grow in a neutral pH, but some are acidophilic, others are alkaliphilic, some are halophilic, and a few are thermophilic [3,20,21].

pH values affect cellular metabolism and secondary metabolite biosynthesis in *Streptomyces* spp. species [22]. The pH was considered adequate for the formation of aerial mycelium close to a neutral value. Ripa et al. [23] reported that extreme pH is unfavorable for secondary metabolite production as in their research, a pH of 8 negatively influenced colony development and mycelial growth.

Silva et al. [24] conducted a similar study to determine the best growth conditions for nigericin production in Czapeck medium, considering certain variables (pH of the medium, medium constituents, fermentation temperature, and fermentation period), where optimal conditions were obtained at a pH of 7 and a fermentation temperature of 30 °C for EUCAL 26 in Czapeck medium, resulting in maximal nigericin production.

By varying the temperature while keeping all other PDA cultivation conditions constant (20 g/L glucose, 2 g/L yeast extract, pH 6.5), it was observed that at 20 °C, incubation of the *Streptosporangium* sp. P1C3 strain took 20 days, without mycelial growth. At 36 °C, the incubation time was reduced to 15 days, but there was still no mycelial growth. According to Bergey [25] and Goodfellow and Fiedler [26], the incubation temperature range for actinobacteria can vary between 25 °C and 37 °C.

Factors such as changes in constituents and pH revealed alterations in the structural characteristics of the *Streptosporangium* sp. P1C3 strain in the four best formulations, based on the data found. These characteristics were best observed in SEM images, as described below: Formulation 01 presented a tangled mass of hyphae with poorly organized spores at the tips, which is considered important for the formation of vegetative aerial mycelium. This formulation allowed sporulation in a smaller number of actinobacteria, resulting in limited aerial mycelium formation.

In contrast, the *Streptosporangium* sp. P1C3 strain grown in experimental formulation 05 showed intense hyphal formation and a large tangle of spores, promoting the formation of aerial mycelium. This indicates that formulation 05 promoted more abundant sporulation of the tested actinobacteria, directly influencing aerial mycelium formation. On the other hand, in formulation 09, abundant production of free and germinative spores was observed, but hyphal appearance was less pronounced compared to other formulations, which directly interfered with the reproduction of aerial hyphae and, consequently, aerial mycelium formation.

Finally, formulation 13 stood out for its abundant formation of aerial hyphae in a spiral structure, allowing visibility of the terminal structures of the hyphae, that is, the aerial mycelium. Additionally, formulation 13 was favourable for sporulation. The commercial PDA formulation exhibited different characteristics compared to the four experimental formulations tested. In the commercial formulation, intense formation of reproductive hyphae was observed, along with a small number of spherical spores and aerial hyphae. However, septations or maturation at the hyphal tips were not observed, which are essential processes for the formation of aerial mycelium, fundamental for pigment production, secondary metabolites, and bioactive compounds, such as potential antimicrobials, antiparasitics, and others.

The different morphologies found in this study corroborate the findings of Maciel [27], who reported that eight isolated actinobacteria, after preliminary SEM analysis, showed different sizes and morphologies. The author reported that it was possible to observe differences in the hyphae and spore chain formation of each isolate and pigment production in different formulations in the culture medium.

Santos et al. [28] reported that the genus *Streptomyces* exhibits a diversity of morphologies, filamentous arrangements, spore forms, and variations in conidial pigmentation. These characteristics are crucial for the identification of isolated species and for understanding the distinct bioactivities associated with each extract produced. Given the versatility of actinobacteria in various fields, it is necessary to research and optimise culture media that can provide better isolation conditions for this group of microorganisms, facilitating research into better or even new possibilities for developing new formulations that are viable for laboratories, as well as meeting the requirements of these bacteria.

According to Pfefferle et al. [29], the composition of the culture medium plays a crucial role in regulating the physiological and biochemical processes of *Streptosporangium*, directly influencing its growth, metabolic activity, and secondary metabolite production. Nutrient availability, carbon and nitrogen sources, pH, temperature, and aeration conditions all interact to modulate enzyme expression, signaling pathways, and biosynthetic gene clusters responsible for bioactive compound synthesis.

Optimising the culture medium can enhance the yield of valuable secondary metabolites, such as antibiotics and antifungal agents, by triggering specific metabolic pathways. Additionally, the presence of precursors or inducers in the medium can activate silent biosynthetic gene clusters, leading to the discovery of novel compounds with pharmaceutical potential. Understanding the intricate relationship between environmental factors and microbial metabolism is essential for maximising bioproduct synthesis and improving industrial fermentation processes.

## 5. Conclusions

The optimisation of the PDA culture medium demonstrated significant improvements in the growth conditions of the bacterium *Streptosporangium* sp. (P1C3), reducing the incubation time and increasing the production of aerial mycelium. Formulations 01, 05, 09 and 13 showed the most promising results, with glucose concentration, yeast extract concentration, pH and temperature directly influencing growth. Response surface analysis identified optimal conditions of 10 g/L glucose, 1 g/L–3 g/L yeast extract, pH levels between 5.7 and 7.2 and temperatures between 24 °C and 32 °C. These findings confirm that PDA optimisation is an effective strategy to improve the isolation of actinobacteria and can contribute to the optimisation of the production of bioactive metabolites with antimicrobial potential by reducing the incubation time in the growth stage of these bacteria.

## Figures and Tables

**Figure 1 microorganisms-13-00654-f001:**
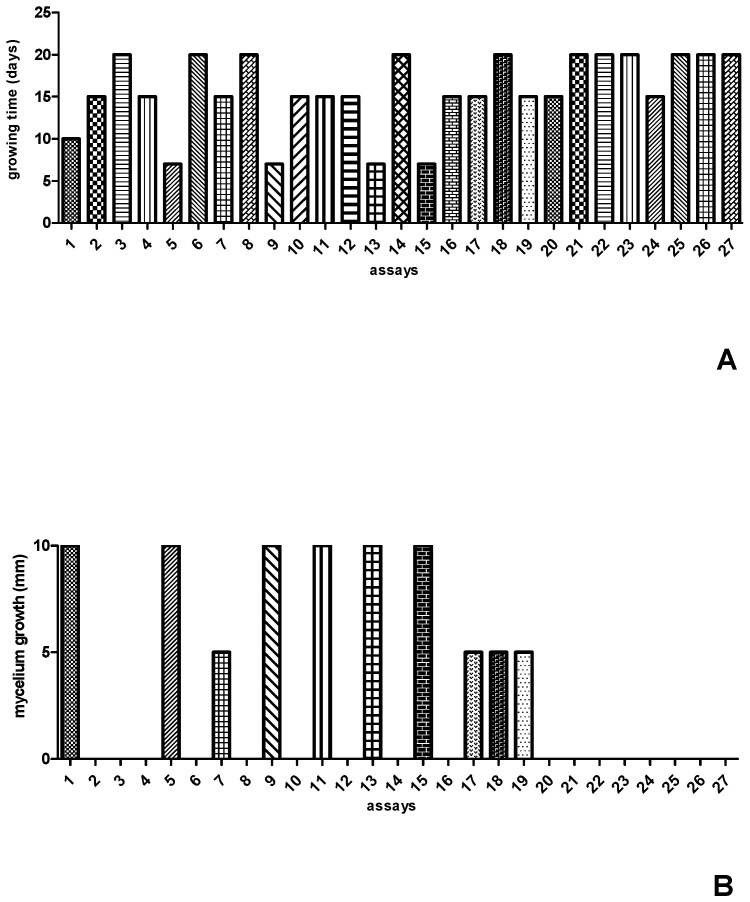
Results of the 27 formulations established in the CCR experimental design for growth time (**A**) and mycelial growth (**B**) of *Streptosporangium* sp.

**Figure 2 microorganisms-13-00654-f002:**
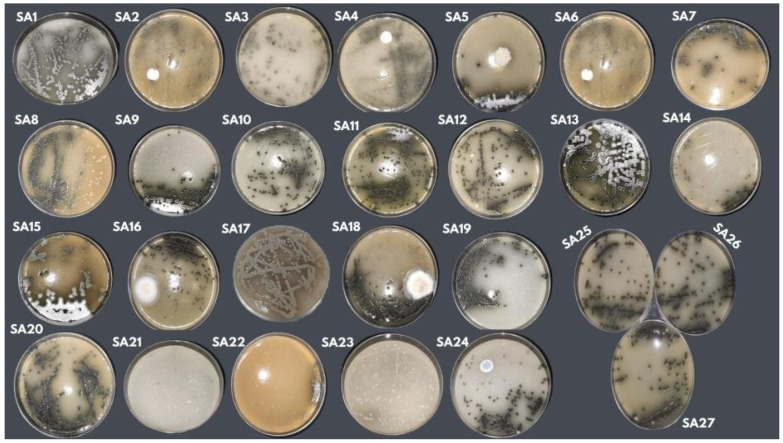
Macromorphological aspect of *Streptosporangium* sp. (P1C3) in PDA culture medium using isolate P1C3 in 27 formulations developed using the CCR experimental design.

**Figure 3 microorganisms-13-00654-f003:**
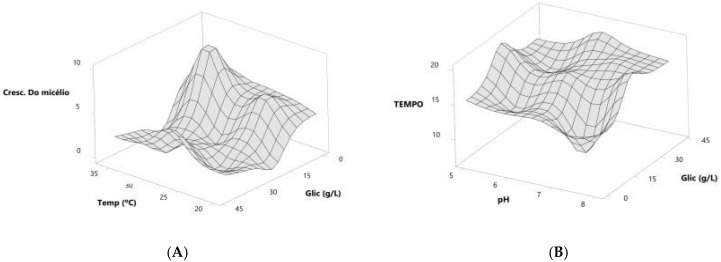
Response surface graphs indicating the trend in ideal values to obtain the best formulations for incubation time (**A**) and aerial mycelial growth (**B**) of the actinobacteria colony.

**Figure 4 microorganisms-13-00654-f004:**
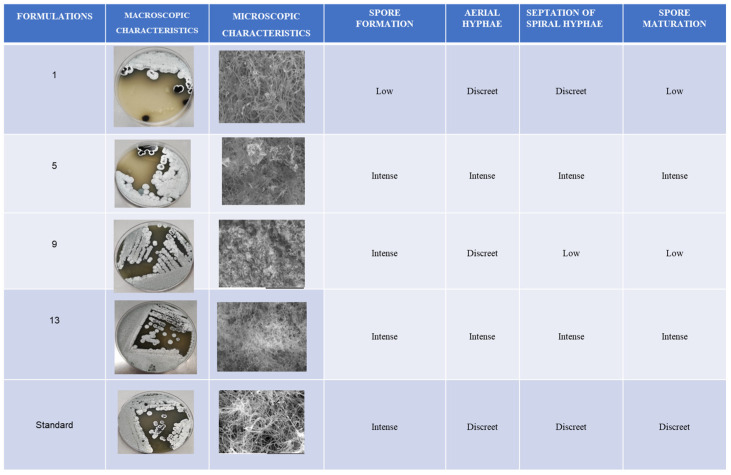
Representation of the macro- and micro-morphological differences presented by the *Streptosporangium* sp. P1C3 strain subjected to SEM.

**Table 1 microorganisms-13-00654-t001:** Experimental matrix containing coded and real values of the independent variables of the central compost rotational (CCR) experimental design.

Formulations	Independent Variables							
	Coded Values				Real Values			
	X_1_	X_2_	X_3_	X_4_	X_1_	X_2_	X_3_	X_4_
1	−1	−1	−1	−1	10	1	5.75	24
2	+1	−1	−1	−1	30	1	5.75	24
3	−1	+1	−1	−1	10	3	5.75	24
4	+1	+1	−1	−1	30	3	5.75	24
5	−1	−1	+1	−1	10	1	7.25	24
6	+1	−1	+1	−1	30	1	7.25	24
7	−1	+1	+1	−1	10	3	7.25	24
8	+1	+1	+1	−1	30	3	7.25	24
9	−1	−1	−1	+1	10	1	5.75	32
10	+1	−1	−1	+1	30	1	5.75	32
11	−1	+1	−1	+1	10	3	5.75	32
12	+1	+1	−1	+1	30	3	5.75	32
13	−1	−1	+1	+1	10	1	7.25	32
14	+1	−1	+1	+1	30	1	7.25	32
15	−1	+1	+1	+1	10	3	7.25	32
16	+1	+1	+1	+1	30	3	7.25	32
17	−2.12	0	0	0	0	02	6.25	28
18	+2.12	0	0	0	40	02	6.25	28
19	0	−2.12	0	0	20	0	6.25	28
20	0	+2.12	0	0	20	04	6.25	28
21	0	0	−2.12	0	20	02	5	28
22	0	0	+2.12	0	20	02	8	28
23	0	0	0	−2.12	20	02	6.5	20
24	0	0	0	+2.12	20	02	6.5	36
25	0	0	0	0	20	02	6.5	28
26	0	0	0	0	20	02	6.5	28
27	0	0	0	0	20	02	6.5	28

X_1_—glucose concentration; X_2_—yeast concentration; X_3_—pH; X_4_—temperature.

**Table 2 microorganisms-13-00654-t002:** Analysis of variance (ANOVA) of the experimental planning results regarding incubation time and mycelium growth.

Parameters	*p* Value	
	Incubations	Micelium Growth
Model	0	0
X_1_	0	0.0015
X_2_	0.0071	0.0082
X_3_	0.0503	0.0047
X_4_	0.0184	0.0025
X_1_X_2_	0.1586	0.0921
X_1_X_3_	0.0075	0.0834
X_1_X_4_	0.0814	0.0034
X_2_X_3_	0.3527	0.6483
X_2_X_4_	0.1553	0.3352
X_3_X_4_	0.2665	0.6912

X_1_—glucose concentration; X_2_—yeast concentration; X_3_—pH; X_4_—temperature

**Table 3 microorganisms-13-00654-t003:** Analysis of the chemical elements in the experimental and commercial formulations, subjected to EDS (energy dispersive spectroscopy).

Formulations	Chemichal Components					
	%					
	C	O	K	P	In	Al
01	65.72 ± 0.1	30.1 ± 0.15	-	-	0.37 ± 0.18	3.90 ± 0.29
05	70.51 ± 0.02	26.71 ± 0.22	1.64 ± 0.27	1.14 ± 0.18	-	-
09	72.09 ± 0.87	24.73 ± 0.48	1.71 ± 0.32	1.46 ± 0.33	-	-
13	70.51 ± 0.75	26.71 ± 0.54	1.64 ± 0.26	1.14 ± 0.75	-	-
Standard	69.86 ± 0.12	28.75 ± 0.30	0.79 ± 0.15	0.60 ± 0.27	-	-

C—carbon; O—oxygen, K—potassium; P—phosphorus; In—indi; Al—Alumin.

## Data Availability

The data presented in this study are openly available in [Caderno Pedagogico] at [10.54033/cadpedv21n11-015], reference number [8].

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
