# Peer review of "Optimisation of Potato Dextrose Agar Culture Medium for Actinobacteria Growth"

_microorganisms, 2025, doi:10.3390/microorganisms13030654_

Round 1

Reviewer 1 Report

Comments and Suggestions for Authors

The current context in microbial research implies a stage of microorganism growth and isolation in laboratory. Thus, it is necessary to optimize culture media and techniques.

Based on this, the manuscript "Optimisation of Potato Dextrose Agar Culture Medium for Actinobacteria Growth" is important for the field of laboratory microbiology.

To the current for of the manuscript, some additional improvements can be performed. 

The Introduction states the necessity of the study and its importance for the field. The aim is clear and well constructed. I suggest the authors to present the hypotheses and/or specific objectives of the study. This will increase the importance of the research.

Materials and methods - this section is explicit and present numerous information necessary for the use of the results and replicate the experiment. I suggest the authors to expand the information related to Central Compost Rotational and add references to this concept. It is interesting and will improve the supplementary information presented in the manuscript.

A multiple comparison test will improve the results obtained by the authors. Standard errors or deviations should be presented to sustain the replicability of the growth phenomena recorded during research. This should be applied to fig 1 and Tab 4.

Results section is explicit and need to be improved with a comparison test. Based on Results the Discussion section is clear and connect the study with international references.

The conclusion section needs to be rewritten to present the main findings of the study, not general observations and perspectives.

Author Response

The current context in microbial research implies a stage of microorganism growth and isolation in laboratory. Thus, it is necessary to optimize culture media and techniques.

Based on this, the manuscript "Optimisation of Potato Dextrose Agar Culture Medium for Actinobacteria Growth" is important for the field of laboratory microbiology.

To the current for of the manuscript, some additional improvements can be performed.

The authors are grateful for the suggestions, following the changes made according to the suggestions of the reviewers.

  1. The Introduction states the necessity of the study and its importance for the field. The aim is clear and well constructed. I suggest the authors to present the hypotheses and/or specific objectives of the study. This will increase the importance of the research.

Answer. Done as suggested

  1. Materials and methods - this section is explicit and present numerous information necessary for the use of the results and replicate the experiment. I suggest the authors to expand the information related to Central Compost Rotational and add references to this concept. It is interesting and will improve the supplementary information presented in the manuscript.

Answer. Done as suggested. Add in line 98 and 114.

  1. A multiple comparison test will improve the results obtained by the authors. Standard errors or deviations should be presented to sustain the replicability of the growth phenomena recorded during research. This should be applied to fig 1 and Tab 4.

Answer. Dear reviewer, the authors thank you for your suggestion. Figure 01 reflects the results of the experimental design that does not require reproduction because it involves combinations based on combinatorial analysis from a mathematical matrix. The number of experiments established in the model plus the central points guarantee the reliability of the process according to the book Planning of Experiments and Process Optimization by Maria Isabel Rodrigues and Antonio Francisco Iemma. Done in Tab. 4

  1. Results section is explicit and need to be improved with a comparison test. Based on Results the Discussion section is clear and connect the study with international references.

Answer. Done as suggested.

  1. The conclusion section needs to be rewritten to present the main findings of the study, not general observations and perspectives.

Answer. Done as suggested

Reviewer 2 Report

Comments and Suggestions for Authors

The reviewed work is an experimental study devoted to the optimization of the nutrient medium for the cultivation of actinobacteria Streptosporagium sp. strain P1C3. The topic of the study is important and relevant, as it contributes to the formation of conditions for the cultivation of producers of biologically active compounds. The aim should specify the specific strain of actinobacteria for which the composition of the nutrient medium was optimized. A broad generalization (actinobacteria) cannot be applied at this stage - for this, growth rates for other species of actinobacteria on optimized formulations should be investigated.

In the Materials and Methods section, the registration number of the strain sequence in the appropriate database should be indicated. Also here and further in the text, the abbreviated spelling of the nutrient medium should be checked - PDA or BDA?

Figure 1 (B) - what are the units of measurement of mycelial growth?

Lines 168-170 - formulas should be placed in the Materials and Methods section.

Figure 3 - In some of the photos of the Petri dishes, there is a reflection of the light source. Is it possible to remove this reflection? Or, you can mark it and write in the notes that it is a reflection of the light source. Otherwise, an inexperienced reader may mistake the reflection for a colony.

Table 3 - Standard - what is the composition?

Line 248 - What are the sources of these elements in 01? Why are they not in the others? Could this have affected the results? This should be discussed

Technical comments are included (see text of the article).

After corrections are made, the article can be published

Author Response

The reviewed work is an experimental study devoted to the optimization of the nutrient medium for the cultivation of actinobacteria Streptosporagium sp. strain P1C3. The topic of the study is important and relevant, as it contributes to the formation of conditions for the cultivation of producers of biologically active compounds..

The authors are grateful for the suggestions, following the changes made according to the suggestions of the reviewers.

  1. The aim should specify the specific strain of actinobacteria for which the composition of the nutrient medium was optimized. A broad generalization (actinobacteria) cannot be applied at this stage - for this, growth rates for other species of actinobacteria on optimized formulations should be investigated.

Answer. Done as suggested

  1. In the Materials and Methods section, the registration number of the strain sequence in the appropriate database should be indicated. Also here and further in the text, the abbreviated spelling of the nutrient medium should be checked - PDA or BDA?

Answer. Done as suggested

  1. Figure 1 (B) - what are the units of measurement of mycelial growth?

Answer. Done as suggested

  1. Lines 168-170 - formulas should be placed in the Materials and Methods section.

Answer. Dear reviewer, thank you for your comment but I would like to clarify that the formula is the result of the analysis of variance and therefore cannot be in the materials and method topic.

  1. Figure 3 - In some of the photos of the Petri dishes, there is a reflection of the light source. Is it possible to remove this reflection? Or, you can mark it and write in the notes that it is a reflection of the light source. Otherwise, an inexperienced reader may mistake the reflection for a colony.

Answer. Done as suggested

  1. Table 3 - Standard - what is the composition?

Answer. Done as suggested

  1. Line 248 - What are the sources of these elements in 01? Why are they not in the others? Could this have affected the results? This should be discussed.

Answer. Done as suggested

P.S. changes are marked in the manuscript in yellow

Reviewer 3 Report

Comments and Suggestions for Authors

Dear Authors,

I have carefully reviewed the manuscript titled Optimization of Potato Dextrose Agar (PDA) Medium for the Growth of Actinobacteria. This study provides valuable insights into improving the cultivation of Streptosporangium sp. (P1C3) through an optimized PDA formulation. By employing a central composite rotational design (CCRD), the research effectively identifies optimal glucose and yeast extract concentrations, pH levels, and temperatures that significantly reduce incubation time and enhance aerial mycelium growth.

While the study makes a significant contribution, further refinement could enhance its scientific depth and broader applicability.

The bibliography used in this manuscript is insufficient and lacks depth, with several key references missing. Additionally, there are sources cited in the text, particularly in lines 59-60, that do not appear in the reference list, raising concerns about the accuracy and completeness of the citations. Given these ambiguities, I strongly recommend that the authors revise the introduction to incorporate more up-to-date and relevant bibliographic sources. A well-supported theoretical framework is essential to strengthen the scientific foundation of the study and ensure the clarity and credibility of the research.

A more detailed analysis of the biochemical and metabolic responses of the strain under different formulations would provide deeper insights into how specific nutrients and physical factors influence actinobacterial growth. Additionally, validating the optimized medium with other actinobacterial strains would strengthen the robustness and generalizability of the findings.

I request that the authors provide details on the vacuum chamber conditions as well as the sample preparation process prior to analysis.

Best regards.

Author Response

I have carefully reviewed the manuscript titled Optimization of Potato Dextrose Agar (PDA) Medium for the Growth of Actinobacteria. This study provides valuable insights into improving the cultivation of Streptosporangium sp. (P1C3) through an optimized PDA formulation. By employing a central composite rotational design (CCRD), the research effectively identifies optimal glucose and yeast extract concentrations, pH levels, and temperatures that significantly reduce incubation time and enhance aerial mycelium growth.

While the study makes a significant contribution, further refinement could enhance its scientific depth and broader applicability.

The authors are grateful for the suggestions, following the changes made according to the suggestions of the reviewers.

  1. The bibliography used in this manuscript is insufficient and lacks depth, with several key references missing. Additionally, there are sources cited in the text, particularly in lines 59-60, that do not appear in the reference list, raising concerns about the accuracy and completeness of the citations. Given these ambiguities, I strongly recommend that the authors revise the introduction to incorporate more up-to-date and relevant bibliographic sources. A well-supported theoretical framework is essential to strengthen the scientific foundation of the study and ensure the clarity and credibility of the research..

Answer. Done as suggested

  1. A more detailed analysis of the biochemical and metabolic responses of the strain under different formulations would provide deeper insights into how specific nutrients and physical factors influence actinobacterial growth. Additionally, validating the optimized medium with other actinobacterial strains would strengthen the robustness and generalizability of the findings.

Answer. Done as suggested. Line 406-420

  1. I request that the authors provide details on the vacuum chamber conditions as well as the sample preparation process prior to analysis.

Answer. Done as suggested. Line 127 - 128

P.S. changes are marked in the manuscript in yellow

Round 2

Reviewer 3 Report

Comments and Suggestions for Authors

Dear Authors,

After reviewing the revised manuscript "Optimization of Potato Dextrose Agar (PDA) Medium for the Growth of Actinobacteria," I recognize the improvements made and the valuable insights provided in this study. Given these enhancements, I find the manuscript suitable for publication and recommend its acceptance.

Best regards,
